# Bacterial Vaginosis and Post-Operative Pelvic Infections

**DOI:** 10.3390/healthcare11091218

**Published:** 2023-04-25

**Authors:** Afroditi Ziogou, Eleftherios Ziogos, Ilias Giannakodimos, Alexios Giannakodimos, Stavros Sifakis, Petros Ioannou, Sotirios Tsiodras

**Affiliations:** 1School of Medicine, National and Kapodistrian University of Athens, 11527 Athens, Greece; 2Department of Gynecology and Obstetrics, University Hospital of Heraklion, 71110 Heraklion, Greece; 3Mitera Maternity Hospital, 71202 Heraklion, Greece; 4School of Medicine, University of Crete, 71003 Heraklion, Greece; 5Fourth Department of Internal Medicine, Attikon General Hospital, 12462 Athens, Greece

**Keywords:** bacterial vaginosis, clindamycin, metronidazole, obstetric complications, pelvic inflammatory disease, preterm delivery, gynecologic complications

## Abstract

Bacterial vaginosis (BV) represents a condition in which the normal protective Lactobacilli, especially those that produce H_2_O_2_, are replaced by high quantities of facultative anaerobes, leading to gynecologic and obstetric post-operative complications. BV is an important cause of obstetric and gynecological adverse sequelae and it could lead to an increased risk of contracting sexually transmitted infections such as gonorrhea, genital herpes, *Chlamydia*, *Trichomonas*, and human immunodeficiency virus. Herein, we reviewed bacterial vaginosis and its association with post-operative pelvic infections. In Obstetrics, BV has been associated with increased risk of preterm delivery, first-trimester miscarriage in women undergoing in vitro fertilization, preterm premature rupture of membranes, chorioamnionitis, amniotic fluid infections, postpartum and postabortal endomyometritis as well as postabortal pelvic inflammatory disease (PID). In gynecology, BV increases the risk of post-hysterectomy infections such as vaginal cuff cellulitis, pelvic cellulitis, pelvic abscess, and PID. BV is often asymptomatic, can resolve spontaneously, and often relapses with or without treatment. The American College of Obstetricians and Gynecologists recommends testing for BV in women having an increased risk for preterm delivery. Women with symptoms should be evaluated and treated. Women with BV undergoing gynecological surgeries must be treated to reduce the frequency of post-operative pelvic infections. Metronidazole and clindamycin are the mainstays of therapy. Currently, there is no consensus on pre-surgery screening for BV; decisions are made on a case-by-case basis.

## 1. Introduction

Bacterial vaginosis (BV) constitutes a gynecological condition characterized by an alteration of the vaginal microenvironment and more specifically an alteration of the normal *Lactobacillus*-dominated vaginal flora, to a flora that includes a variety of facultative and obligatory anaerobic bacteria; this alteration may be associated with adverse outcomes following a gynecological/obstetrical surgical intervention, such as an increased risk of post-operative infections after pelvic surgery [1]. The vaginal flora responsible for BV can either be transient or of a more permanent nature and consists of various microorganisms including *Gardnerella vaginalis*, *Prevotella* spp., *Porphyromonas* spp., *Bacteroides* spp., *Peptostreptococcus* spp., *Mycoplasma hominis*, *Ureaplasma urealyticum*, *Mobiluncus* spp., *Fusobacterium* spp., *Sneathia* spp., *Atopobium vaginae*, and *Clostridium* spp. [2,3]. A polymicrobial biofilm on the epithelial cells of the vagina is a characteristic feature of BV [4].

Concerning BV in pregnant women, it has been related to an increased risk of preclinical pregnancy loss and miscarriage in women undergoing in vitro fertilization (IVF) during the first semester [5,6]. BV can result in the development of various infections commonly seen in everyday practice, such as chorioamnionitis and amniotic fluid contamination, postpartum and postabortal endomyometritis, endometrial bacterial colonization, preterm delivery, postpartum fever, as well as postabortal PID [5,7]. In addition, the presence of BV may facilitate the development of an infection associated with other sexually transmitted bacteria and viruses, such as *Chlamydia trachomatis*, *Trichomonas vaginalis*, *Neisseria gonorrheae*, human papillomavirus, and herpes simplex virus (HSV) [8,9,10,11]. Women diagnosed with BV are at a higher risk of recurrence [2,12,13,14]. HSV-2 seropositivity is associated with BV while effective suppressive therapy with valacyclovir did not decrease the risk [8,15]. Importantly, BV constitutes an important risk factor for the acquisition and transmission of HIV [16,17,18], since patients with BV presented with an almost six-fold increased quantity of HIV shed in vaginal secretions compared to those without BV [19,20,21,22,23]. BV may cause endocervical inflammation presenting as mucopurulent cervicitis, while it has been associated with a three-fold higher risk of vaginal cuff cellulitis and abscess development after vaginal hysterectomy [24,25,26,27]. Furthermore, BV is also related to a sixfold increased incidence of post-cesarean endomyometritis, after adjusting for various factors, such as labor duration, time after membrane rupture, and maternal age [28]. Nevertheless, and despite the aforementioned associations, in the majority of cases, BV remains asymptomatic, mainly occurs during the menstrual cycle, and can resolve spontaneously with or without treatment [29,30].

The consequences of post-operative infections caused by BV include the extended use of antibiotics, the need for further surgical interventions, prolongation of hospital stay, and the need for readmission in affected patients, and are additionally associated with elevated hospitalization costs [31,32]. The current recommendations on the management of BV focus on applying screening tests and treatment modalities in order to reduce the frequency of post-operative pelvic infections. The aim of the present effort was to comprehensively review the current literature on the association between bacterial vaginosis and post-operative pelvic infections occurring after gynecological/obstetrical surgeries.

## 2. Epidemiology, Risk Factors, and Pathophysiology of BV

BV constitutes the most common cause of abnormal vaginal discharge in women of childbearing age, accounting for 40–50% of such cases [33,34]. During pregnancy, the prevalence of BV varies from 5.8% to 19.3% while variability in prevalence exists between different races and/or ethnicities [6]. It remains unclear whether these findings reflect genetic, socioeconomic, or behavioral discrepancies. In a combined study including 1461 pregnant asymptomatic women, the prevalence of BV rate was 12.3% [33]. BV occurs in 33–36% of women attending sexually transmitted disease clinics, and up to 25% of those attending gynecologic clinics [35].

Of note, Royce et al. identified that the differences in the vaginal flora of pregnant women may depend on ethnicity [36]. They compared black and white women during the third trimester of pregnancy and observed that, in total, 22.3% of black and 8.5% of white women developed BV [36]. In a cohort study from 2003 to 2013 in the USA including 12,340 mother–infant pairs, with 2468 being exposed to BV and 9872 being unexposed, BV-exposed mothers were more likely to be Hispanic or Black [37].

Risk factors for BV mainly include hormonal changes, current or prior use of specific medication such as antibiotics and immunosuppressants, and the existence of foreign bodies, such as cloth or toilet tissue in the vagina [38]. On the other hand, the appearance of BV has not been associated with the presence of comorbidities such as diabetes or immunosuppression [39]. BV is not considered a sexually transmitted infection, however, it has been associated with smoking as well as women’s hygiene behavior, i.e., vaginal douching [40,41] and sexual habits, such as the number of male sexual partners [41,42], female partners [43], new sexual partners, age of first sexual encounter, number and frequency of sexual contacts, lack of condom use [41,44], use of an intrauterine device (IUD), especially copper-containing [34,45], as well as infection with HIV and other sexually transmitted bacteria [41,46]. Male genitalia may harbor BV-associated bacteria [47,48] while circumcision is associated with reduced risk [49]. On the other hand, therapeutic interventions targeting the male sex partner do not prevent BV recurrence in the affected female partner [50]. Studies evaluating therapeutic interventions among women having sex with women are currently lacking. Contraception using hormonal regimens does not appear to increase the risk and may actually prevent its occurrence [51,52]. Black race and low socioeconomic status constitute other risk factors [53]. Interestingly, compared to women without BV, the sexual commencement of those with BV occurs at an earlier median age [41,43]. The prevalence of the disease is related to the menstrual cycle [54,55]. A large cross-sectional study conducted among 53,652 rural married women in China reported that menstrual cycles of more than 35 days, less than 3 days of menstruation, dysmenorrhea, and use of an intrauterine device were associated with BV [56,57]. In a study from Kenya, it was noted that BV prevalence decreased with increasing women’s age [58].

In a cross-sectional study of parous women in the USA, the effects of psychosocial stress, household income, and neighborhood socioeconomic parameters on the risk of BV were investigated. Having a low income was associated with an increased prevalence of BV among African American women in a statistically significant manner, but this was not the case for White American women. Furthermore, more stressful life events were significantly associated with higher BV prevalence among both African American and White American women. Moreover, neighborhood socioeconomic status was associated with the increased BV prevalence univariately by principal components analysis among White American women, but this was not to be found significant after adjusting for individual-level risk factors [59]. Interestingly, in another cross-sectional study conducted in Tanzania, the primary education level or below was found to be a significant predictor of BV along with an age of less than 30 years, vaginal douching, HIV infection, sexually transmitted infection, early age of initiation of sexual activity, and having more than one sexual partner in their lifetime [41]. Another study reporting data from more than 2 million Swedish women aged 15–50 years from national registry data showed that women with a lower education level had a 46% higher risk of BV, which was reduced by about 25% when adjusting for other co-variates [60]. Moreover, in a study from 2003 to 2013 in the USA including 12,340 mother–infant pairs, with 2468 being exposed to BV and 9872 being unexposed, BV-exposed mothers were less likely to have had a college degree compared to the BV-unexposed mothers [37].

Although various risk factors and pathogenetic mechanisms related to the development of BV have been described, its exact etiology remains unknown. The healthy vaginal flora is inhabited by a variety of *Lactobacillus* spp. (90–95% of total bacteria), such as *L. chrispatus*, *L. iners*, *L. jensenii*, *L. vaginalis*, and *L. gasseri* that maintain a low pH (<4.5), produce bacteriostatic and bactericidal substances and, as a result, impede the occurrence of infection [15]. A decrease in or absence of *Lactobacillus* spp. provokes an increase in the vaginal pH that leads to an overgrowth of anaerobic Gram-negative rods, leading to the development of BV [24]. More specifically, *Lactobacillus* spp. produces several chemical byproducts, such as lactic acid, bacteriocins, and hydrogen peroxide, that decrease the vaginal pH ≤ 4.5 and create an unfavorable environment for facultative pathogens [61]. This alteration in the concentration of *Lactobacillus* and pH value may facilitate an overgrowth of anaerobes, producing larger amounts of proteolytic carboxylase enzyme [62]. This can dissect vaginal peptides into a variety of amines that are volatile, malodorous, and associated with increased vaginal transduction and squamous epithelial cell exfoliation [62]. Interestingly, the mucinase and sialidase levels of vaginal fluid were considerably higher in women with BV compared to women with normal vaginal flora [62]. According to a conceptual model for the pathogenesis of BV, developed by Schwebkeet et al., *G. vaginalis* is the dominant pathogen, while other pathogens mainly act synergistically as secondary intruders [63]. *Lactobacilli* may have detrimental effects on the occurrence of other infections such as trichomoniasis [64]. On the other hand, an extravaginal reservoir of vaginal bacteria may serve as an important risk factor for the recurrent episodes of BV [1]. Importantly, the pathophysiology of BV does not include signs of inflammation, and this is the rationale for using the term ‘vaginosis’ rather than the term ‘vaginitis’ since the principal event leading to the clinical symptoms and signs seem to be a result of bacterial dysbiosis due to microbial imbalance in the microbiota of the vagina [65].

Since the prevalence of BV is not the same in different patient populations despite having similar exposures, and due to the higher incidence in patients of African descent, genetic studies have been performed and revealed specific genetic loci implying a role for biological pathways related to cell signaling and mucosal immunity in the pathogenesis of BV [66]. For example, a genome-wide associated study that included women from Kenya identified an association of genes encoding for molecules involved in innate immunity, such as Toll-like receptors (TLRs), interleukin-8, TIRAP, MYD88, with specific microorganisms involved in the pathogenesis of BV [67]. Another study in HIV-infected adolescents identified an association between specific TLR single-nucleotide polymorphisms and BV, while similar studies also exist in non-HIV-infected patients [68,69].

Additionally, there are studies suggesting an important role for metabolomics in the pathogenesis of BV. For example, a relatively recent study showed important differences in the composition and concentrations of metabolites in patients with and without BV [70]. More specifically, when a comparison of the metabolite profiles in cervicovaginal lavage from patients with and without BV was performed, levels of 62% in 279 metabolites were significantly different in women with BV. In particular, women with BV had lower levels of amino acids and dipeptides, as well as higher levels of amino-acid catabolites, polyamines, and of the eicosanoid 12-hydroxyeicosatetraenoic acid which is a known biomarker for inflammation [70]. In another study aiming to identify bacterial and metabolic hallmarks for BV, increased concentrations of *Prevotella*, *Atopobium*, and *M. hominis* were more prevalent in the vaginal fluid of women with BV. The proton nuclear magnetic resonance of vaginal fluid in women with BV and without BV also identified and quantified 17 previously unreported molecules. Changes in the levels of amines, amino acids, organic acids, monosaccharides, short-chain fatty acids, and nitrogenous bases were associated with BV. More specifically, kynurenine, maltose, and NAD(+) were the most characteristic of the non-BV status, while malonate, nicotinate, and acetate were characteristic of BV [70].

## 3. Bacterial Vaginosis: Diagnosis

Along with colposcopy and microscopic examination, history serves as the mainstay of diagnosis. The Amsel criteria primarily based on microscopic findings are used and require the presence of at least three of the four following parameters: (a) thin, grayish/white, homogeneous discharge; (b) pH of vaginal fluid >4.5; (c) detection of fishy odor upon addition of KOH to vaginal fluid (positive white test); and (d) the presence of significant clue cells (defined as >20% of the total vaginal epithelial cells seen on 100× magnification on saline microscopy) [71]. Normal pH (<4.5) excludes the diagnosis of BV. The presence of cervical mucus, blood, or sperm in the vaginal secretions can increase the pH. Notably, levels of pH above 4.5 need further investigation, and a differential diagnosis between BV, trichomoniasis, and mucosal purulent cervicitis is required since all these entities can be associated with elevated levels of pH in the alkaline side.

Although the Gram stain constitutes the gold standard for the diagnosis of BV [72], its usage is time-consuming and demands more resources and expertise compared to Amsel criteria [73]; the detection of three Amsel criteria correlates well with Gram stain findings [74]. However, when the Gram stain is used for the diagnosis of BV, the sensitivity and specificity of Amsel criteria are over 90% and 77%, respectively [75]. The Gram-stained smear is examined using Nugent criteria or Hay/Ison criteria and its sensitivity varies from 62 to 100% [76]. Nugent criteria are used to quantify or grade *Lactobacilli*, *Bacteroides*/*Gardnerella*, and *Mobiluncus* in order to create a scale of flora deviation, varying from normal (score = 0–3), to intermediate (score = 4–6) and frank (score = 7–10) BV [61]. Of note, the Papanicolaou smear is considered unreliable for the diagnosis of BV, with a sensitivity of 49% and a specificity of 93% [77]. Finally, the amino test is characterized by high specificity (up to 90%) and low sensitivity [75].

Vaginal cultures of *G. vaginalis* play no role in the diagnosis of BV since it is not specific to the entity and BV constitutes a polymicrobial infection; associated bacteria can be cultured even in asymptomatic women. Similarly, a cervical Pap test is of no use. Although positive cultures for *G. vaginalis* are observed in the majority of symptomatic patients, this bacterium can be detected in up to 50–60% of healthy asymptomatic women, constituting its isolation not being diagnostic for BV [62]. Multiple molecular tests as well as point-of-care tests have been made available to clinicians to facilitate diagnosis at the bedside [78]. A relatively recent study compared the DNA hybridization test Affirm VPIII and the Gram stain using the Nugent criteria in diagnosing BV. Out of 115 positive vaginal specimens for BV as diagnosed by Gram stain, the Affirm VPIII test identified the existence of *G. vaginalis* in 107 (93%) [79]. Symptomatic women with pH changes and the presence of amine odor appear to benefit the most from this test. In another study, the combination of a positive DNA probe and vaginal pH of more than 4.5 presented a sensitivity of 95% and a specificity of 99%, respectively [80]. The OSOM BV Blue system constitutes a chromogenic diagnostic test that depends on the activity of sialidase enzymes in vaginal fluids [79,81]. This diagnostic modality carries a sensitivity between 88 and 94% as well as a specificity between 91 and 98% when compared with Amsel and Nugent criteria [82]. The FemExam Test Card Uses the presence of the *G. vaginalis* byproduct trimethylamine, proline aminopeptidase, and vaginal pH with a sensitivity of 91% with a lower specificity of 61%, which has been evaluated in syndromic management in resource-poor settings [83]. With regard to molecular testing, it has been associated with superior sensitivity and specificity in detecting bacterial DNA from pathogens that have been identified are highly associated with the diagnosis of BV (i.e., *G. vaginalis*, *A. vaginae*, BVAB2, or *Megasphaera* type 1) and various species of *Lactobacilli* (i.e., *L. crispatus*, *L. jensenii*, and *L. gasseri*) [78,83,84]. It can even be performed on self-collected specimens. Various quantitative multiplex PCR assays with high sensitivity and specificity are available [85,86,87]. Cultures and/or rapid nucleic acid testing may be helpful in differentiating BV from other infectious entities such as chlamydia or gonorrhea.

## 4. Obstetrical Complications Associated with BV 

Pregnant women with BV—especially those with long-standing or untreated disease—present with a higher risk for preterm delivery; BV constitutes a risk factor for plasma-cell endomyometritis, postpartum fever, endometrial bacterial colonization, and postabortal infections [88,89,90]. The exact prevalence of post-cesarean infections worldwide is not well delineated, ranging from 2.5% to 20.5% [58]. Post-cesarean infections mainly include wound infection and endomyometritis. Wound infection may present with discharge, erythema, and the induration of the incision, though it usually complicates in 2–7% of patients and generally develops 4–7 days after cesarean delivery [91]. Table 1 summarizes the studies reporting on obstetric complications associated with BV.

Surgical site infection (SSI) constitutes a common post-cesarean complication that requires antibiotic therapy and drainage for symptom resolution. Other authors noted that wound infection affects 11% of these women [92]. Of note, a post-cesarean wound infection identified prior to hospital discharge will lead to the prolongation of hospital stay, elevated hospitalization costs, and increased need for readmission [93].

The presence of BV during pregnancy is associated with a two-fold higher risk for preterm delivery [94]. BV concomitant with African American race is strongly associated with preterm birth [29]. Other risk factors associated with preterm birth and BV include nulliparity, young age, smoking, low educational attainment, low socio-economic status, and sexually transmitted diseases [94]. Ten percent of all births are preterm and are a major cause of intraventricular hemorrhage, acute respiratory illnesses, and neurodevelopmental disturbance. The prevalence and severity of these outcomes from preterm delivery are higher with earlier gestational age.

Hiller et al. demonstrated that women with BV are 40% more likely to have a preterm birth, low-birth-weight infant compared to women without BV [89]. The presence of BV at an early gestational age is associated with preterm delivery [95]. Various studies confirm that women with BV are more vulnerable to ascending genital tract infection; amniotic fluid infection results mostly from an invasion of lower genital tract bacteria through the placental membranes [96,97]. Frequently, most isolates recovered from the amniotic fluid of women with intact membranes are the microorganisms associated with BV and pregnant with BV are twice as likely to have an invasion of the amniotic fluid in comparison to women with *Lactobacillus* predominant vaginal flora [94]. An increased risk of intraamniotic infection among women with BV at less than 34 weeks of gestation was reported by Hitti and Hillier et al. Ascending genital tract can induce preterm labor by the production of pro-inflammatory cytokines such as IL-1, IL-1β, and TNF [98]. Cytokines were found in higher concentrations in the amniotic fluid in women with spontaneous preterm delivery due to infection [98]. High concentrations of IL-8 in the vagina and anaerobic flora were both associated with amniotic fluid infection [99,100]. Furthermore, Watts et al., in another study, observed that women with BV at the time of cesarean section were 18% more likely to have an infection compared to those with normal vaginal flora (22% vs. 4%) [28]. In a multivariate analysis that included various factors related to postpartum endomyometritis, BV was associated with a nearly six-fold increased risk (OR 5.8, 95% CI 3.0–10.9) in women with extended duration of labor, or prolonged duration of membrane rupture and advanced maternal age. Postpartum endomyometritis was diagnosed in 34/97 (35%) pregnant women with BV compared to 35/365 (10%) of those with normal vaginal flora (*p* < 0.001), despite using antibiotic prophylaxis [28]. Another very recent study in Egypt evaluating a total of 200 pregnant women prepared for elective emergency cesarean section, with half of them having BV and half of them not, identified that cesarean wound infection was more frequent among those who had BV, occurring in up to 57% before discharge among patients with BV, with an eight-fold increased risk among this patient group (RR 8.0, 95% CI 1.02–62.79).

Hillier et al. proposed a link between BV, histological chorioamnionitis, and microorganism detection from the chorioamnion [100].

Hay et al. studied the Gram-stained vaginal smears of 718 pregnant women who were examined until 36 weeks of pregnancy. Among the pregnant women who initially had normal vaginal flora, only 2.4% had developed BV until 36 weeks of gestation [101]. Among 32 women with BV initially, half had abnormal vaginal flora until 36 weeks [101]. Pregnant women who developed BV before 20 weeks of gestation were at higher risk for preterm delivery compared to those who developed it after 20 weeks [101]. 

In a meta-analysis that evaluated the effect of BV in preterm birth in studies published from 2008 to 2022, the correlation of BV with preterm birth was solidly confirmed with a relative risk of 1.44 (95% CI 1.19–1.73) [102]. Another study in more than 3000 adult pregnant female patients in Denmark identified BV to be independently associated with the preterm birth of a low-birth-weight infant (OR 2.5, 95% CI 1.6–3.9), indicated preterm delivery (OR 2.4, 95% CI 1.4–4.1), low birth weight (OR 1.95, 95% CI 1.3–2.9) and clinical chorioamnionitis (OR 2.7, 95% CI 1.4–5.1). These results were derived from a multivariate regression analysis where an adjustment for previous preterm delivery, smoking, previous miscarriage, previous conization, gestational diabetes, fetal death, and preterm premature rupture of membranes was performed [90].

Importantly, BV was also shown to be associated with adverse outcomes among full-term infants. In a cohort study from 2003 to 2013 in the USA including 12,340 mother–infant pairs, with 2468 being exposed to BV and 9872 being unexposed, following adjustment, BV was associated with an increased risk of respiratory distress and assisted ventilation at birth (aRR = 1.28, 95% CI 1.02–1.61), admission to the NICU (aRR = 1.42, 95% CI 1.11–1.82), and neonatal sepsis (aRR = 1.60, 95% CI 1.13–2.27) among infants who were full-term. Among preterm infants, exposure to BV was only associated with a higher risk for admissions to the NICU (aRR = 1.24, 95% CI 1.04–1.46) [37].

In a study evaluating the role of the newly identified fastidious BV-associated bacteria *Sneathia* (*Leptotrichia*) *sanguinegens*, *Sneathia amnionii*, *Atopobium vaginae*, and BV-associated bacteria 1 (BVAB1) in 545 women, an endometrial biopsy was obtained for histology at baseline and at 30 days, and showed that the persistent detection of bacteria associated with BV was common (range 58% for *A. vaginae* to 82% for BVAB1), and this was associated with an elevated risk for persistent endometritis (RR 8.5, 95% CI 1.6–44.6) at 30 days. In models adjusted for gonorrhea and chlamydia, endometrial BV-associated bacteria were shown to be associated with recurrent PID (RR 4.7, 95% CI 1.7–12.8), while women who were found to be positive for these bacteria in the cervix and/or endometrium were at higher risk of the development of infertility (RR 3.4, 95% CI 1.1–10.4) [103].

BV is also found to be correlated with plasma-cell endometritis, endometrial bacterial colonization, and postpartum endomyometritis [88,89]. Postpartum endometritis refers to a polymicrobial infection of the decidua, characterized by fever, uterine tenderness, and purple discharge from the uterus. Postpartum endometritis is one of the most common complicating infections, which can follow an aggressive course progressing into endomyometritis, up to pelvic abscess or even generalized peritonitis and septicemia. Postpartum endometritis complicates in 2–16% of women who underwent cesarean delivery. Culture-based methods have shown that early postpartum endometritis is characterized by the presence of multiple microbiotas often associated with BV including *Gardnerella vaginalis*, *Peptococcus* spp., *Bacteroides* spp., *Staphylococcus epidermidis*, *Streptococcus agalactiae* and *Ureaplasma urealyticum* [104]. Plasma cell endometritis was frequently present in women with BV and without other vaginal or cervical infections. Korn et al. [105], evaluated 41 women with endometrial biopsies who presented vaginal discharge or pelvic pain. Among these women, 22 were diagnosed with BV by the Gram-stain examination of vaginal fluid. Among the women with BV, 10/22 had plasma cell endometritis compared to 1/19 control (OR 15, 95% CI 2–686; *p* = 0.004). BV-associated microorganisms were cultured from the endometria of 9/11 women and 8/30 women without plasma cell endometritis (OR 12.4, 95% CI 2–132; *p* = 0.02). Treating pregnant women for at least 24 weeks of gestation, Pitt et al. evaluated the topical metronidazole or placebo prior to cesarean section [106]. The results of this trial were that 7% of the traced pregnant patients developed post-cesarean endometritis versus 17% of patients who received a placebo (*p* = 0.04). Alexander et al. in a randomized trial used a regimen of two doses of 2 g metronidazole in pregnant women with BV between 16 and 24 weeks of pregnancy. They did not find a decrease in the rates of preterm delivery or perinatal outcomes among women randomized to receive metronidazole [107].

Secondary infertility appears to be unusual in these patients. A severe infection may result in exosalpingitis, but endosalpingitis is uncommon [108]. Bacteremia due to *G. vaginalis* may be observed but blood cultures should be collected in blood culture media without the anticoagulant that is toxic to the microorganism.

The principal focus of BV screening has been to minimize the risk of preterm delivery. Pregnant women who are diagnosed with BV before 20 weeks of gestation were at higher risk for preterm delivery compared to those who developed BV after 20 weeks [109]. 

The therapeutic management of women with symptomatic BV late in pregnancy not only aims to treat their symptoms but additionally reduces the risk of postpartum endomyometritis. On the contrary, the therapeutic approach of asymptomatic pregnant patients is not well defined in the literature. Due to the lack of data concerning the cost-effectiveness of screening in asymptomatic women with BV, the identification and treatment of these patients can be extremely challenging [29].

The United States Preventive Services Task Force suggests against screening for BV in pregnant women who are not at higher risk for preterm delivery and stated that the current evidence is not adequate to access the balance of benefits and harms of screening for BV in pregnant women that are at increased risk of preterm delivery [29].

## 5. Late Miscarriage 

BV acts as a risk factor for progression to the upper genital tract: active chlamydia infection increases the risk of postabortal PID, by three to four times [110]. Additionally, BV was shown by some authors to increase the risk of postabortal PID by 100% [29]. A higher rate of late miscarriage (13–23 weeks’ gestation) was demonstrated in women with BV compared to those without BV. 

Hay et al. [101] and Mac Gregor et al. [111] found increased RRs of 3.9 and 3.1 for miscarriage in the mid-trimester, respectively, when BV was diagnosed before 16 weeks of gestation. Ralph et al. studied the relationship between BV and conception and miscarriage in the first trimester [112]. Among 237 women, 61 (32.1%) were diagnosed with BV, among which 22/61 (36.1%) miscarried during the first 13 weeks of gestation. A significantly increased risk of miscarriage was found in women with BV compared to those with normal vaginal flora (RR 1.95, 95% CIs 1.11–3.42). This risk was increased after adjustment: increasing maternal age, smoking, and history of three or more miscarriages, no previous live birth, and polycystic ovaries (RR 2.49, 95% CIs 1.21–5.12).

Ugwumadu et al. identified a three-fold increase in miscarriage risk during the first trimester, but in other studies, BV is found to be related to late miscarriage compared with the first-trimester pregnancy loss [113].

## 6. Post-Operative Infections after Gynecological Surgeries Associated with BV

In non-pregnant women, BV has been linked to vaginal cuff cellulitis following abdominal hysterectomy, cuff abscess, and pelvic abscess [24,25,29]. A pelvic abscess occurs in less than 1% of women subjected to obstetric or gynecologic surgical procedures [58]; it develops when pelvic cellulitis or pelvic hematoma expands into the parametrial soft tissue [114]. Four studies were available concerning the risk of post-operative pelvic infections following abdominal hysterectomy among women with BV (Table 2). The first study by Soper et al. included 161 women scheduled for abdominal hysterectomy for a benign condition [25]. Women with BV have are at increased risk (RR: 3.2, 95% CI 1.5–6.7) for vaginal cuff cellulitis compared to those without BV [25]. They found that women with BV had a three times higher risk of developing cuff cellulitis, cuff abscess, or both following abdominal hysterectomy compared to women without BV. Moreover, microorganisms associated with BV including *G. vaginalis*, *Bacteroides* spp., or *Peptostreptococcus* spp. were isolated from the vaginal cuff of more than 60% of cases of cuff cellulitis.

The second study by Larson et al. included 70 women undergoing abdominal hysterectomy for benign conditions excluding postmenopausal women, and it indicated a four-fold higher chance (RR: 4.1 95% CI 1.4–13.3) for the development of post-operative pelvic infection in women with BV compared to those without [115]. Out of twenty, seven women with BV (35%), characterized by the presence of clue cells in the vaginal discharge, developed post-hysterectomy infections versus 4/50 (8%) women without BV [115].

A nationwide study from Sweden demonstrated a 3-fold higher risk (RR 3, 95% CI 1.3–7) of post-operative infections after abdominal hysterectomy for benign conditions among women with BV compared to those without BV [116]. Finally, Lin et al. remarked on an increased rate of post-operative infections after a major gynecologic surgery in women with BV [117]. In this study, 175 women underwent major gynecologic surgery. These women were evaluated for the presence of BV based on Nugent’s criteria. Thirty-six percent of the positive BV women developed a post-operative fever, compared with 20% of the *Lactobacillus*-predominant group of women and 12% of the intermediate-BV group (*p* = 0.017). The difference between the positive-BV group and the intermediate-BV group was statistically significant (*p* = 0.045 and *p* = 0.007, respectively), while the difference between the intermediate-BV group and the *Lactobacillus*-predominant group was not.

Women who were diagnosed with an intermediate flora were not at higher risk of post-operative infections [117]. This could be due to the fact that they did not exclude postmenopausal women. Postmenopausal women lack *Lactobacilli* in the vaginal flora and will score as intermediate flora according to the Nugent criteria. An earlier study showed that postmenopausal women had a lower rate of post-operative infection in comparison to younger women [116]. 

Regarding intervention with antimicrobials, Larsson et al. noted that women with BV or intermediate flora benefit from antibiotic treatment and should be preoperatively initiated on treatment [115]. Pre- and post-operative treatment for at least four days with metronidazole rectally considerably reduces vaginal cuff cellulitis among women with BV. Among 59 women diagnosed with abnormal vaginal flora, post-operative infections were not noted in the treated arm compared to 27% in the untreated arm. Treatment administration also decreased the vaginal cuff cellulitis rate from 9.5 to 2% among 83 women with *Lactobacillus*-dominated flora. Treatment had no impact on the rate of the wound infection [118].

## 7. Postabortal Endometritis/PID

The frequency of postabortal PID after a first-trimester surgical abortion is 4–12% and post-operative pelvic cellulitis occurs in up to 70% of patients undergoing hysterectomy without prophylaxis [119].

Larson et al. noted a higher incidence of postabortal PID after first-trimester surgical abortion in women with BV [120]. A double-blind randomized, multicenter study enrolled 231 women who were randomized to receive either 500 mg three times daily for 10 days or a placebo. Among 174 women who could be evaluated, postabortal PID developed in 14 after the abortion. Postabortal PID developed in 3/174 (3.6%) in the treatment group compared to 11/174 (12.2%) in the placebo group (*p* < 0.05). Moreover, the treatment of BV decreases the incidence of postabortal PID by 8.7% (95% CI, 0.9–16.6); resulting in a threefold reduction in the rate of postabortal PID [120]. In another study by Larson et al., women with *Mobiluncus* and clue cells in the vaginal discharge were found to have a higher risk of postabortal PID [121]; 531 women underwent a first-trimester surgical abortion and 16/531 women were excluded from the study. *Mobiluncus* was identified in 81 (15.2%), *C. trachomatis* in 39 (7.6%), and in 11 cases, both *Mobiluncus* and *C. trachomatis* were detected. In women with *Mobiluncus*, the incidence of postabortal PID was 8/74 (10.8%) compared to 18/398 (4.5%) women with neither *C. trachomatis* nor *Mobiluncus* [121]. Additionally, clue cells were present in 65/198 (32.8%) Gram-stained smears, while 133/198 (67.2%) showed normal epithelial cells. In the two groups, the incidence of postabortal PID was 11.8% and 3.2%, respectively (*p* = 0.01). Postabortal PID includes an infection caused by microorganisms ascending to the endometrium, fallopian tube, and/or contiguous structure. Pelvic inflammatory disease (PID) includes any combination of endometritis, salpingitis, pelvic peritonitis, or tubo-ovarian abscess. However, endogenous microorganisms that are part of the lower genital tract flora can also be recovered from the endometrium, fallopian tube, and peritoneal fluid of women with acute PID. BV has been associated with PID or histologic endometritis in three studies [105]. Endometrial cultures from women with PID before the initiation of antibiotic therapy show a correlation with microorganisms implicated in BV. Two studies reported the recovery of BV-associated microorganisms from more than half of the women diagnosed with PID [105]. However, in another study, BV-associated microorganisms made-up only 11% of the endometrial isolates of women with PID [122]. Hillier et al. studied 178 women with suspected PID with endometrial biopsies for histologic and microbiological study; 85 of them underwent laparoscopy to diagnose salpingitis [123]. Histologic endometritis was confirmed in 117 (65%) women. Among the women who underwent laparoscopy, salpingitis was present in 68, while 23% of these women did not have endometritis. Some, but not all microorganisms associated with BV coexisted with endometritis. Paavonen et al. reported that 9/31 (29%) women with histologic endometritis had BV compared with none of the 14 women without endometritis (*p* = 0.02) [124]. Korn et al. evaluated 41 women without PID, *N. gonorrheae*, or *C. trachomatis* and found that 10/22 women with BV had histologic endometritis compared to 1/19 control subjects [105]. The diagnosis of endometritis in these studies was based on the detection of plasma cells in the endometrial stroma, whereas in Hillier’s study, the diagnosis was based on the detection of plasma cells in the stroma plus polymorphonuclear leucocytes in the endometrial epithelium.

In a randomized double-blind placebo-controlled trial that enrolled 273 women with BV that underwent a surgical termination of pregnancy, the efficacy of metronidazole administration in those with BV was studied [110]. These women were randomized to either receive a 2 g metronidazole suppository or an identical placebo perioperatively. Intention to treat analysis showed that the incidence of post-operative upper genital tract infections was 8.5% and 16% after the administration of metronidazole and placebo, respectively, suggesting that treatment with metronidazole may decrease the risk of postabortal infections [110].

In a retrospective longitudinal follow-up study of 4945 abortions, Carlsson et al. studied postabortal infections; 17.5% of these patients underwent surgical abortions. Of all the women who tested positive for BV at the screening and therefore received antibiotics, only 1.4% developed a postabortal infection [125]. Among women who were negative at the screening, 1.7% developed infectious complications [125]. In another study by Charonis and Larsson in 2006 that included patients with BV, after medical abortions, a rate of infectious complications of 2.4% was noted, which was lower than the one after surgical abortions, i.e., at 4.5%. Importantly, in that study, there were no cases of postabortion PID among participants who were treated for BV after diagnosis with the QuickVue Advanced pH and Amines test with microscopic confirmation [126]. In another study from Sweden and Norway, the infection prevalence was 4.8% following a surgical abortion; these patients had not received antibiotics [36].

Multiple studies have indicated that the preoperative treatment of BV has been associated with a decreased rate of postabortal infections; the WHO recommends antibiotic prophylaxis administration to prevent infectious morbidity associated with abortions [38,39,46]. When using the “screen and treat method”, all patients subjected to surgical termination procedures will be tested for several bacteria. Antibiotic treatment can be administered before, during, or after the surgical abortion. Prior research showed that the timing of antibiotic administration relative to the time of abortion does not influence the rate of postabortal infections [126].

McElligot et al., in a cost-comparison study, evaluated the options of all BV patients’ treatment resulting in a cuff infection rate of 4% with a mean cost of USD 593 [127]. The diagnostic method: “test all patients for BV and treat if they are positive”, also proved inexpensive, having a mean cost of USD 623 and a 4.2% cuff infection rate. This model suggests that metronidazole administration lowers the infectious complications following major gynecologic surgery. 

## 8. Bacterial Vaginosis: Screening and Treatment

Treatment should be considered in women with symptoms. Since BV may resolve spontaneously in up to one-third of non-pregnant and one-half of pregnant women, the screening and treatment of BV remain controversial [29,128]. However, asymptomatic pregnant women with previous preterm delivery may benefit from treatment. Thus, even though screening could be performed in this population, the USPSTF clearly recommends against screening pregnant persons who are not at an increased risk for preterm delivery [29]. Beyond pregnant women, there are very few studies providing data regarding the cost-effectiveness of screening in asymptomatic women with BV, thus, it is unclear whether screening should be recommended in specific patient populations who might be at a higher risk for complications, as those who will undergo gynecological surgery, as described above, since another acceptable option could be to “treat all patients”, rather than “test and treat if positive” [29,127]. The decision for screening in gynecological cases should be taken on a case-by-case basis.

Some benefits related to the treatment of BV include the reduction in sexually transmitted infections, relief of associated symptoms, and reduction in the risk for post-operative infections [16,129,130]. The oral or intravaginal administration of metronidazole or clindamycin to symptomatic non-pregnant women results in a high rate of clinical cure in approximately 70–80% at four weeks [131]. The recent CDC guidance recommends an oral regimen of metronidazole 500 mg twice daily for seven days [132]. However, whether multi-day oral/vaginal administration is more efficient than single-daily dosage has not been determined to date, so a multi-day regimen administration is usually the preferred choice [133]. Despite the common misconception, there is currently no convincing evidence for the development of disulfiram-like interaction between alcohol and metronidazole [134]. Refraining from sexual activity or condom use is advised during BV treatment while douching is not encouraged [129].

Multiday vaginal treatment consists of 5 g of metronidazole in gel form (0.75%) once daily for five days [118]. Furthermore, the regimen of choice for clindamycin is a seven day-course of 2% clindamycin cream (5 g, one full applicator) intravaginally. Intravaginal clindamycin treatment has been related to a higher incidence of clindamycin-resistant anaerobic bacteria. Pseudomembranous colitis has also been associated with the oral and topical use of clindamycin. However, clindamycin administration may be less effective than the metronidazole regimen; thus, vaginal clindamycin cream remains an alternative therapeutic choice [132]. Oral clindamycin is administered in a dose of 300 mg twice daily for seven days, while an alternative approach suggests the intravaginal administration of clindamycin ovules 100 mg once daily at bedtime for three days [132,133]. If neither metronidazole nor clindamycin is available nor tolerated, the administration of tinidazole (2 g orally once daily for two days) or secnidazole (2 g oral granules in a single dose) would constitute suitable oral alternatives. Tinidazole is a second-generation nitroimidazole presenting with a longer half-life (12–14 h vs. 7 h for metronidazole), while it also has fewer side effects compared to metronidazole [132,135]. Randomized trials have proven that it is at least as efficient as metronidazole, without superiority and a single-dose regimen seems to be as effective as vaginal clindamycin cream [118]. Additionally, secnidazole is a nitroimidazole antibiotic with a longer half-life than metronidazole (approximately 17 h vs. 8 h) [122]. A single oral dose of 2 g secnidazole was at least as efficient as a course of 500 mg metronidazole administered orally twice daily for seven days [53]. 

All pregnant women with symptomatic BV should be treated, while treatment is indicated for asymptomatic women who are scheduled for gynecologic surgical procedures. Oral treatment is effective and has not been linked to adverse fetal or obstetric side effects. The therapeutic options include metronidazole 500 mg orally twice daily for seven days; metronidazole 250 mg orally three times daily for seven days; clindamycin 300 mg orally twice a day for seven days; intravaginal metronidazole gel 5 g once a day for five days; or intravaginal clindamycin cream 5 g at bedtime for seven days [129].

A meta-analysis of randomized trials performed in obstetrics population has proven that the treatment of asymptomatic BV does not decrease the prevalence of preterm delivery [136]. In a 2013 Cochrane meta-analysis including 21 trials with 7847 pregnant women presenting with BV, antibiotic therapy was highly effective in eradicating infection but did not considerably reduce the rate of preterm delivery prior to 37 weeks (OR 0.88, 95% CI 0.77–1.09) nor the risk of the preterm premature rupture of membranes (OR 0.74, 95% CI 0.30–1.84) [137]. 

Recent research is evaluating the use of BV biofilm-disrupting agents such as TOL-463 in an attempt to prevent relapses and achieve a definite cure [138], while no studies support the use of probiotics [139,140,141,142,143]. Follow-up is unnecessary if there is a resolution of symptoms and retreatment with the same or a different regimen may be used in women with recurrence [129,144]. For multiple recurrences, long-term chemoprophylaxis has been used [145,146,147,148]. The treatment of the partner does not appear to affect the response and is not recommended [50]; nevertheless, this remains an issue for further study with some preliminary efforts reporting positive results [149].

## 9. Discussion

The present narrative review summarizes the risk factors, etiology, pathophysiology, and diagnostic criteria, as well as the adverse consequences of BV in obstetrics and gynecology. For the conduction of the present review, studies on bacterial vaginosis providing clinical data and outcomes on obstetrics and gynecology with a focus on post-operative infections were included. More specifically, a PubMed search was performed with the terms ‘bacterial vaginosis’ alone and in combination with ‘obstetric complications’, ‘post-operative complications’, ‘outcomes’, and ‘pathophysiology’ and all the resulting original studies published until April 2023 were screened by the authors. Moreover, the reference of those studies were also searched for additional relevant articles, while narrative and systematic reviews on the topic of bacterial vaginosis were also retrieved and studied, and their references were also searched for the addition of more relevant studies in the present review.

BV is a gynecological condition that emerges due to an alteration of the vaginal microbiome and more specifically because of a reduction in the normal *Lactobacillus*-dominated vaginal flora, and the increase in facultative and obligatory anaerobic bacteria. BV is the most common cause of abnormal vaginal discharge in women of childbearing age, accounting for 40–50% of such cases [33,34]. Risk factors for BV mainly include ethnicity, since women of Black and Hispanic origin may present with BV more often [36,37], as well as hormonal changes, current or prior use of specific medications such as antibiotics and immunosuppressants, and the existence of foreign bodies [38]. BV has been associated with smoking as well as women’s hygiene behavior, i.e., vaginal douching [40,41] and sexual habits, such as the number of male sexual partners [41,42], female partners [43], new sexual partners, age of first sexual encounter [41,43], number and frequency of sexual contacts, lack of condom use [41,44], use of an IUD [34,45], as well as infection with HIV and other sexually transmitted diseases [41,46]. Low socioeconomic status constitutes another risk factor [53]. Social and economic factors may also influence the risk of BV. For example, low income has been associated with an increased prevalence of BV, while more stressful life events were significantly associated with higher BV. Moreover, the neighborhood socioeconomic status was also associated with increased BV prevalence [59]. Finally, a lower education level is also associated with a higher risk of BV [37,41].

Even though, in the majority of cases, BV remains asymptomatic and can resolve spontaneously with or without treatment [29,30], in pregnant women, it has been linked with an increased risk of preclinical pregnancy loss and miscarriage in women undergoing IVF during the first semester [5,6]. Pregnant women with BV, especially those with long-standing or untreated disease, present with a higher risk for preterm delivery; BV constitutes a risk factor for plasma-cell endomyometritis, postpartum fever, endometrial bacterial colonization, and postabortal infections [88,89,90].

Furthermore, BV can lead to the development of several commonly seen infections, such as chorioamnionitis, postpartum and postabortal endomyometritis, preterm delivery, postpartum fever, and postabortal PID [5,7]. Moreover, BV can increase the risk of the transmission of sexually transmitted infections [8,9,10,11]. BV may cause endocervical inflammation presenting as mucopurulent cervicitis and has been associated with a three-fold higher risk of vaginal cuff cellulitis and abscess development or pelvic infection after vaginal hysterectomy [24,25,26,27]. Additionally, a four-fold higher chance for the development of post-operative pelvic infection in premenopausal women with BV undergoing abdominal hysterectomy was noted [115]. Additionally, women with BV undergoing major gynecological surgery were also more likely to develop post-operative fever and infections [116]. Importantly, BV is also associated with adverse outcomes among full-term infants, as a link between BV in the mother and an increased risk of respiratory distress and assisted ventilation, admission to the NICU, and neonatal sepsis among infants who were full-term has been documented. BV may also increase the risk of postabortal PID [29], while in women with BV, a higher rate of late miscarriage (13–23 weeks’ gestation) has been demonstrated.

BV spontaneously resolves in up to one-third of non-pregnant and one-half of pregnant women and, thus, the screening and treatment of BV remain controversial [29,128]. However, asymptomatic pregnant women with previous preterm delivery may benefit from treatment [29]. All pregnant women with symptomatic BV should be treated, while treatment is indicated for asymptomatic women who are scheduled for gynecological surgical procedures. Figure 1 graphically depicts the development of BV and its possible consequences that were presented herein.

This review has some notable limitations. First of all, it is a narrative review. This means that, contrary to a systematic review that has strict and formal requirements about the literature search and criteria regarding study inclusion and exclusion, this process was performed less strictly herein. Thus, the data provided by the current review may not be exhaustive, even though they were also selected after an adequate literature search and a careful consideration of the available evidence. Furthermore, the majority of studies included herein are relatively old, however, there is a relative paucity of data in the last two decades on the topic of BV, thus, these studies most likely reflect the best available evidence regarding BV. Finally, due to language restrictions, only studies in English were included in the analysis, and thus, particularities of BV in populations described in other languages, may not be represented in the current review.

## 10. Conclusions

The treatment of BV may prevent post-operative infections after major gynecological surgery or surgical termination of pregnancy. Women undergoing hysterectomy, the surgical termination of pregnancy, and cesarean delivery should be screened for BV, and those who are positive should be treated. Asymptomatic pregnant women with previous preterm delivery may benefit from treatment, but the screening and treatment of these women remain controversial. Metronidazole and clindamycin remain the mainstay of treatment for BV.

## Figures and Tables

**Figure 1 healthcare-11-01218-f001:**
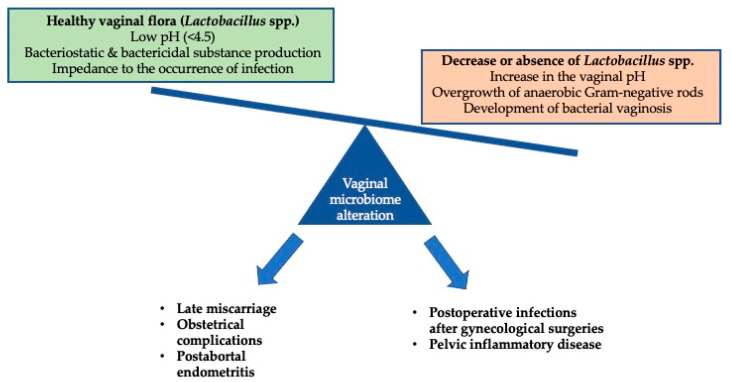
Brief summary of the pathophysiology and consequences of bacterial vaginosis.

**Table 1 healthcare-11-01218-t001:** Obstetric complications associated with BV.

Study	Type of Surgery	N	RR, 95% CI	Outcome	Comments
Korn et al., 1995	Endometrial biopsies in women with vaginal discharge or pelvic pain	41	OR: 15, 95% CI 2–686	Plasma cell endometritis	10/22 women with BV had plasma cell endometritis vs. 1/19 controls 9/11 women cultured microorganisms associated BV and 8/50 without plasma cell endometritis
Ralph et al., 1999	Miscarriage in the first trimester	237 61/237 (32.1%) women with BV	RR: 1.95 95% CI 1.11–3.42 RR: 2.49 95% CI 1.21–5.12	22/61 (36.1%) miscarried during the first 13 weeks of gestation	The risk is increased after adjustment: increasing maternal age, smoking, history of 3 or more miscarriages, no previous live birth, PCO
Watts et al., 1990	Cesarean section	97	OR: 5.0 95% CI 3.0–10.9	Postpartum endometritis	34/97 (35%) women with BV developed postpartum endometritis vs. 36/365 (10%) without BV (*p* < 0.001)
Pitt et al., 2001	Cesarean section (treating pregnant women for at least 24 weeks with metronidazole or placebo)	112		Postpartum endometritis	7% of women treated with topical metronidazole developed postpartum endometritis vs. 17% of patients who received a placebo (*p* = 0.04)
Alkady et al., 2022	Cesarean section	200 (100 with BV 100 without BV)	RR: 8.0, 95% CI 1.02–62.79	Cesarean wound infection within one week	Prospective study on 200 women prepared for cesarean section in Egypt
Jacobsson et al., 2002		924 Prevalence of BV was 15.6%	RR: 3.26, 95% CI 1.38–7.71 RR: 2.10 95% CI 0.90–4.91	Postpartum endometritis Preterm delivery	In this trial, the possible association between BV with premature delivery and postpartum endometritis was studied
Silver et al., 1989	Amniocentesis and aspiration of amniotic fluid	125		Intraamniotic infection	32/125 (26%) with BV 22/32 (69%) developed intraamniotic infection vs. 43/93 (46%) had intraamniotic inf without BV *p* = 0.03
Hillier et al., 1996	Endometrial biopsies of 178 women, 85 with laparoscopy	178	RR: 2.6, 95% CI 1.1–5.7 for anaerobic Gram-negative associated BV	Pelvic inflammatory disease	117/178 (65%) women with histologic endometritis 58/85 (69%) with salpingitis and 23% of these without endometritis
Haggerty et al., 2016	Endometrial biopsies at baseline and at 30 days	545	RR 8.5, 95% CI 1.6 to 44.6 RR 4.7, 95% CI 1.7 to 12.8 RR 3.4, 95% CI 1.1 to 10.4	Risk for persistent endometritis Recurrent PID Infertility	A study examining the role of recently identified fastidious BV-associated bacteria *Sneathia* (*Leptotrichia*) *sanguinegens*, *Sneathia amnionii*, *Atopobium vaginae*, and BV-associated bacteria 1 (BVAB1)
Svare et al., 2006		3262 singleton pregnant women Prevalence of BV was 16%	OR 1.95, 95% CI 1.3–2.9 OR 2.5, 95% CI 1.6–3.9 OR 2.4, 95% CI 1.4–4.1 OR 2.7, 95% CI 1.4–5.1	Low birth weight Preterm delivery of a low-birth-weight infant Indicated preterm delivery Clinical chorioamnionitis	Multivariate regression analysis adjusted for previous miscarriage, previous preterm delivery, previous conization, smoking, gestational diabetes, fetal death, and preterm premature rupture of membranes

BV: bacterial vaginosis; CI: confidence intervals; OR: odds risk; PCO: polycystic ovaries; RR: relative risk.

**Table 2 healthcare-11-01218-t002:** Post0operative infections following gynecological surgeries associated with BV.

Study	Type of Surgery	N	RR, 95% CI	Outcomes	Comments
Soper et al., 1990	Abdominal hysterectomy	161	RR: 3.2, 95% CI 1.5–6.7	Cuff cellulitis, cuff abscess	×3 higher risk of post-operative infections
Larson et al., 1991	Abdominal hysterectomy	70 7/20 (35%) with BV 4/50 (8%) without BV	RR: 4.1 95% CI 1.4–13.3	Vaginal cuff cellulitis	No antibiotic prophylaxis
Person E et al., 1996	Abdominal hysterectomy	1060	RR: 3 95% CI 1.3–7	Post-operative pelvic infections	
Lin et al., 1999	Major gynecologic surgery	175 36% of pos BV vs. 20% without BV and 12% with an intermediate-BV group (*p* = 0.017)		Post-operative fever (*p* = 0.045)	
Larsson et al., 1992	Surgical abortion in the first trimester	231 Randomized to receive 500 mg metronidazole ×3 daily or placebo		14/174 women developed PID 3 in metronidazole group (3.84%) 11 (12.2%) in the placebo group (*p* < 0.05)	Treated BV reduces the post-operative infection rate more than 3 times
Larsson et al., 1989	Surgical abortion in the first trimester	515 65/198(32.8%) with clue cells 37/65(56.9%) with *Mobiluncus*	*p* = 0.01 *p* = 0.015	7/59 (11.8%) with clue cells developed postabortal PID 8/74 (10.8%) with *Mobiluncus* developed postabortal PID	

BV: bacterial vaginosis; CI: confidence intervals; PID: pelvic inflammatory disease; RR: relative risk.

## Data Availability

Not applicable.

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
