# Peer review of "Bacterial Vaginosis and Post-Operative Pelvic Infections"

_healthcare, 2023, doi:10.3390/healthcare11091218_

Round 1

Reviewer 1 Report

The manuscript is a review on bacterial vaginosis and postoperative pelvic infections after gynecological surgeries. However, I believe that it should be improved as follows: 

Section 2. Authors briefly describe the epidemiology of BV. It would be interesting to detail the association with social status and education grade. Also, please insert more references to each sentence to strength your statement.

 Section 3. This section could be tightened to the Section 2 since they overlap. As also stated in the section 2, the information brought by the authors did not lead to a solid conclusion.

Section 4. To this reviewer, this section is irrelevant to the manuscript and should be removed.

 Section 5. Please, address typing issues at line 195 and revised throughout the text.

 Table 1. Are there only published papers from almost 20 years ago ? Authors should insert more recent data in the review.

To this reviewer, the main text should be rewritten since it lacks cohesion. Authors seemed to just mention data without connecting the information. The manuscript as a review should be fluid and clear to summarize the information to the reader without losses. Authors should insert a summary figure with the main idea of the manuscript.

Author Response

The manuscript is a review on bacterial vaginosis and postoperative pelvic infections after gynecological surgeries. However, I believe that it should be improved as follows: 

Section 2. Authors briefly describe the epidemiology of BV. It would be interesting to detail the association with social status and education grade. Also, please insert more references to each sentence to strength your statement.

Response: Thanks for the comment. Indeed, there seems to be an association between BV and social status and education level. Thus, we found some representative studies and added that information in the second section, as can be seen in the revised version of the manuscript. Furthermore, we added more references in sentences of that section that only had one reference as can be seen in the same section of the revised manuscript.

Section 3. This section could be tightened to the Section 2 since they overlap. As also stated in the section 2, the information brought by the authors did not lead to a solid conclusion.

Response: Thanks for the comment. As can be seen in the revised version of the manuscript, we merged the two sections (2 and 3) and made some modifications that make this new section easier for the reader. More specifically, we added a sentence that somehow summarizes the pathophysiology of BV that has to do with alteration of the vaginal microbiota and added a reference for that.

Section 4. To this reviewer, this section is irrelevant to the manuscript and should be removed.

Response: Thanks for the comment. We are very skeptical about that. Even though the aim of the present review was not to focus on the diagnosis of BV, we feel that this information should be included so the reader could know how the diagnosis is made. Since no other reviewer made a negative comment about this section, we wanted to consider keeping it, unless, if another round of revisions is performed, the reviewer insists on removing it, or the editor agrees with the reviewer to delete it.

Section 5. Please, address typing issues at line 195 and revised throughout the text.

Response: We corrected that. The manuscript was also corrected by a native English speaker and minor corrections were performed.

Table 1. Are there only published papers from almost 20 years ago ? Authors should insert more recent data in the review.

Response: Thanks for the comment. We have searched the literature again and eventually found a very small number of newer studies that are on the topic of our review. We have added them both in the text and in the tables, as can be seen in the revised version of the manuscript. However, the studies that involve bacterial vaginosis – a relatively underappreciated topic in gynecology – are mostly older, thus, the original version of the review had already included the most important studies on the topic. For example, even other recent literature reviews on the topic also have relatively old references, as can be seen, for example, in the following review: https://doi.org/10.1016/j.ajog.2019.09.002.

To this reviewer, the main text should be rewritten since it lacks cohesion. Authors seemed to just mention data without connecting the information. The manuscript as a review should be fluid and clear to summarize the information to the reader without losses. Authors should insert a summary figure with the main idea of the manuscript.

Response: Thanks for the comment. We agree that there is a lot of information in this review. And we also agree that having a lot to say may make things look complicated. Thus, taking into consideration this comment as well as another comment from another reviewer, we created a discussion section at the end of the revised version of this manuscript to summarize the manuscript. Furthermore, we created a figure that summarizes in a simple and comprehensible manner the main message this manuscript tries to convey. This figure can be seen at the end of the discussion section of the revised version of the manuscript.

Reviewer 2 Report

Dear the editor-in-chief,

This review study brings a comprehensive, well-organized and up-to-dated set of knowledge in case of “Bacterial Vaginosis and Postoperative Pelvic Infections”. I do recommend this study for publication in your journal. However some minor comments are given in the attached file which is suggested to be addressed by the authors.

Kind regard

Author Response

Dear the editor-in-chief,

This review study brings a comprehensive, well-organized and up-to-dated set of knowledge in case of “Bacterial Vaginosis and Postoperative Pelvic Infections”. I do recommend this study for publication in your journal. However some minor comments are given in the attached file which is suggested to be addressed by the authors.

Response: Thanks for the comments. We have added a point-by-point response to the pdf file that was provided in the electronic system and revised the manuscript accordingly as can be seen in the revised version.

Reviewer 3 Report

The manuscript titled “Bacterial Vaginosis and Postoperative Pelvic Infections” represents a relevant and interesting review of the current literature on the association between bacterial vaginosis and postoperative pelvic infections occurring after gynecological/obstetrical surgeries.

The entire manuscript is more or less well-written in all its parts. However, there are some minor obstacles that need to be revised before the manuscript is ready for publication.

These, among others. include:

Line 100…The term IUD- should be followed by its full name in the parenthesis.

Line 145…The term PH should be replaced by pH.

Table 1 should be mentioned in the manuscript text.

Furthermore, some parts of the manuscript should be better complemented with corresponding references.

For example: 

Line 167-174… The authors have written: “ A relatively recent study compared the DNA hybridization test Affirm VPIII and the Gram stain using the Nugent criteria in diagnosing BV. Out of 115 positive vaginal specimens for BV as diagnosed by Gram stain, the Affirm VPIII test identified the existence of G. vaginalis in  (93%). Symptomatic women with pH changes and the presence of amine odor appear to benefit the most from this test. In another study, the combination of a positive DNA probe and vaginal pH of more than 4.5 presented with a sensitivity of 95% and specificity of 99%, respectively.” The references are missing for these statements.

Line 203-204… The authors have written: “Bacterial vaginosis concomitant with African-American race is 203 strongly associated with preterm birth.” The reference is missing.

Line 215-218… The authors have written: “Various studies confirm that women with BV are more vulnerable to the ascending genital tract infection; amniotic fluid infection results mostly from an invasion of lower genital tract bacteria through the placental membranes. Frequently, most isolates recovered from the amniotic fluid of women with intact membranes, are the microorganisms associated with BV, and pregnant with BV are twice as likely to have an invasion of the amniotic fluid as women with Lactobacillus predominant vaginal flora. An increased risk of intraamniotic infection among women with BV at less than 34 weeks of gestation has been reported by Hitti and Hillier et al. Ascending genital tract can induce preterm labor by production of pro-inflammatory cytokines such as IL-1, IL-1β, and TNF. Cytokines were found in higher concentrations in the amniotic fluid in women with spontaneous preterm delivery due to infection.” The references are missing.

The same applies to the text: “Furthermore, Watts et al., in another study, observed that women with BV at the time of cesarean section were 18% more likely to have an infection compared to those with normal vaginal flora (22% vs 4%).” In this case, the corresponding reference (ref no. 28) is cited in line 235. However, it will be appropriate also to cite the corresponding reference at the end of the sentence mentioning the authors of the research.

The same applies for the references no. 84 and 106.

Line 355… The authors have written: “In a study, Larson et al noted a higher incidence of postabortal PID after first-trimester surgical abortion in women with BV [101].” The term- in a study- is not necessary.

A minor revision and acceptance of the manuscript are suggested.

Author Response

The manuscript titled “Bacterial Vaginosis and Postoperative Pelvic Infections” represents a relevant and interesting review of the current literature on the association between bacterial vaginosis and postoperative pelvic infections occurring after gynecological/obstetrical surgeries.

The entire manuscript is more or less well-written in all its parts. However, there are some minor obstacles that need to be revised before the manuscript is ready for publication.

These, among others. include:

 Line 100…The term IUD- should be followed by its full name in the parenthesis.

Response: Thanks for the comment. The full name of the term IUD- is added to the sentence in the parenthesis.

Line 145…The term PH should be replaced by pH.

Response: Thank you for the comment. The term PH has been replaced by pH in line 145.

Table 1 should be mentioned in the manuscript text.

Response: Thanks for the comment. Table 1 has been mentioned in the manuscript as can be seen in the revised version.

Furthermore, some parts of the manuscript should be better complemented with corresponding references.

 For example: 

Line 167-174… The authors have written: “ A relatively recent study compared the DNA hybridization test Affirm VPIII and the Gram stain using the Nugent criteria in diagnosing BV. Out of 115 positive vaginal specimens for BV as diagnosed by Gram stain, the Affirm VPIII test identified the existence of G. vaginalis in  (93%). Symptomatic women with pH changes and the presence of amine odor appear to benefit the most from this test. In another study, the combination of a positive DNA probe and vaginal pH of more than 4.5 presented with a sensitivity of 95% and specificity of 99%, respectively.” The references are missing for these statements.

Response: Thanks. We have added references to that specific point.

Line 203-204… The authors have written: “Bacterial vaginosis concomitant with African-American race is 203 strongly associated with preterm birth.” The reference is missing.

Response: Thanks. We have added references to that specific point.

Line 215-218… The authors have written: “Various studies confirm that women with BV are more vulnerable to the ascending genital tract infection; amniotic fluid infection results mostly from an invasion of lower genital tract bacteria through the placental membranes. Frequently, most isolates recovered from the amniotic fluid of women with intact membranes, are the microorganisms associated with BV, and pregnant with BV are twice as likely to have an invasion of the amniotic fluid as women with Lactobacillus predominant vaginal flora. An increased risk of intraamniotic infection among women with BV at less than 34 weeks of gestation has been reported by Hitti and Hillier et al. Ascending genital tract can induce preterm labor by production of pro-inflammatory cytokines such as IL-1, IL-1β, and TNF. Cytokines were found in higher concentrations in the amniotic fluid in women with spontaneous preterm delivery due to infection.” The references are missing.

Response: Thanks. We have added references to that specific point.

The same applies to the text: “Furthermore, Watts et al., in another study, observed that women with BV at the time of cesarean section were 18% more likely to have an infection compared to those with normal vaginal flora (22% vs 4%).” In this case, the corresponding reference (ref no. 28) is cited in line 235. However, it will be appropriate also to cite the corresponding reference at the end of the sentence mentioning the authors of the research.

Response: Thank you for your comment. The corresponding reference has been cited as can be seen in the corresponding part of the revised version of the manuscript.

The same applies for the references no. 84 and 106.

Response: Thank you for your comment. The requested changes have been performed. However, the numbering of the references has changed due to addition of new references in the revised version of the manuscript.

Line 355… The authors have written: “In a study, Larson et al noted a higher incidence of postabortal PID after first-trimester surgical abortion in women with BV [101].” The term- in a study- is not necessary.

Response: Thank you for your observation. The term “in a study” has been removed from the sentence in question.

A minor revision and acceptance of the manuscript are suggested.

Response: Thank you for your comment.

Reviewer 4 Report

The review addresses the effect of microbiota changes in induction of BV and associated OB/GYN implications. Overall, the review is well structured. I only have minor comments.

line 71-72: "The consequences of postoperative infections caused by BV beyond the extended use of antibiotics...." this sentence is not clear, what exactly do you mean by "beyond the extended use of antibiotics".

line 94: "existence of foreign bodies..." Please elaborate what do you mean by foreign bodies, it is a bit vague. Do you mean pathogenic bacteria? Viruses? if so which bacteria/virus? Or do you mean some other endotoxin factors etc.?

line 219-221: "and pregnant with BV are twice as likely to have an invasion of the amniotic fluid as women with Lactobacillus predominant vaginal flora", do you mean "in comparison to" women with Lactobacillus predominant vaginal flora instead of "as"? I thought Lactobacillus predominant vaginal flora is more healthy like and supposed to reduce BV associated pathologies. Please verify.

Minor language errors in line 113-114, please rectify. 

Author Response

The review addresses the effect of microbiota changes in induction of BV and associated OB/GYN implications. Overall, the review is well structured. I only have minor comments.

line 71-72: "The consequences of postoperative infections caused by BV beyond the extended use of antibiotics...." this sentence is not clear, what exactly do you mean by "beyond the extended use of antibiotics".

Response: Thank you for your comment. In order to clarify its meaning, the sentence The consequences of postoperative infections… hospitalization costs” has been modified to “The consequences of postoperative infections caused by BV include the extended use of antibiotics, the need for further surgical interventions, prolongation of hospital stay and the need of readmission in affected patients and are associated with elevated hospitalization costs”, as can be seen in the corresponding part in the revised version of the manuscript.

line 94: "existence of foreign bodies..." Please elaborate what do you mean by foreign bodies, it is a bit vague. Do you mean pathogenic bacteria? Viruses? if so which bacteria/virus? Or do you mean some other endotoxin factors etc.?

Response: Thank you for your observation. The following examples have been added to the sentence in line 94 “such as cloth or toilet tissue” to further elaborate the meaning of foreign bodies.

line 219-221: "and pregnant with BV are twice as likely to have an invasion of the amniotic fluid as women with Lactobacillus predominant vaginal flora", do you mean "in comparison to" women with Lactobacillus predominant vaginal flora instead of "as"? I thought Lactobacillus predominant vaginal flora is more healthy like and supposed to reduce BV associated pathologies. Please verify.

Response: Thanks for the comment. Indeed we mean in comparison to women with Lactobacillus predominant vaginal flora. The term “as” has been replaced by the term “in comparison to” in lines 219-221.

Minor language errors in line 113-114, please rectify. 

Response: Thank you for the comment. The sentence “In a study from Kenya 113 noted that BV prevalence decreased with increasing age of the women” has been modified to “In a study from Kenya it was noted that BV prevalence decreased with increasing women’s age”. 

Round 2

Reviewer 1 Report

The authors have revised the manuscript and addressed most of issues. However, there are more questions that may improve the review:

1. Regarding the Etiology, there are molecular alterations that could favor BV ? There are several publications showing metabolic signature in BV condition. To insert more detailed information regarding molecular, metabolic ang gene expression alterations would improve the manuscript.

2. In the Discussion section, authors state that this is a narrative and not a systematic review. Please, insert clearly the criteria for paper selection.

Author Response

The authors have revised the manuscript and addressed most of issues. However, there are more questions that may improve the review:

  1. Regarding the Etiology, there are molecular alterations that could favor BV ? There are several publications showing metabolic signature in BV condition. To insert more detailed information regarding molecular, metabolic ang gene expression alterations would improve the manuscript.

Response: Thanks for the interesting comment. Indeed, there is literature on this topic. Thus, we have modified the manuscript by adding to big paragraphs at the end of section two, to allow the reader to see the pathogenetic mechanisms that are associated with metabolomics and the genetic background of the women. We have added 5 relatively recent references with studies focusing on this subject, such as studies evaluating the effect of SNPs, genome-wide association studies, as well as studies evaluating the levels of specific molecules/metabolic parameters, that prove an association between these factors and the development of BV.

  1. In the Discussion section, authors state that this is a narrative and not a systematic review. Please, insert clearly the criteria for paper selection.

Response: Thanks for the comment. We had mentioned previously that this is not a systematic review, since systematic reviews have very strict methodologies about study search and inclusion, while narrative reviews do not. However, according to the reviewer’s comment, we added some descriptive information about how the study search was performed in the present review. This information can be seen in the first paragraph of the discussion section. More specifically, for the conduction of the present review, studies on bacterial vaginosis providing clinical data and outcomes on obstetrics and gynecology, with a focus on post-operative infections were included. More specifically, a Pubmed search was performed with the terms ‘bacterial vaginosis’ alone and in combination with ‘obstetric complications’, ‘post-operative complications’, ‘outcomes’, and ‘pathophysiology’ and all the resulting original studies published until April 2023 were screened by the authors. Moreover, the reference of those studies were also searched for additional relevant articles, while, narrative and systematic reviews on the topic of bacterial vaginosis were also retrieved and studied, and their references were also searched for addition of more relevant studies in the present review.

For more details please see the revised version manuscript.